# Limitations of calculating theoretical solutions for closed BCMP queueing networks and verification of alternative theoretical values by parallel simulation

**Shinya Mizuno** ⓘ *, **Haruka Ohba**

Juntendo University, Chiba, Japan

* s.mizuno.kc@juntendo.ac.jp, s.mzn.eng@gmail.com

**Data Availability Statement:** The datasets generated during and/or analysed during the current study are available in the GitHub repository

## Abstract

This study applied a closed BCMP queueing network to a real-world model, examining the limitations of the theoretical solution and the possibility of replacing theoretical values with those from parallel simulation. Parallel computing was applied to mean value analysis (MVA). We first obtained computational and theoretical values by varying the number of nodes from 33 to 300 and customers from 250 to 1500 in a system with three customer classes. The computation time increased proportionally with the number of nodes but exponentially with the number of customers, reaching 146,798.86 seconds for 33 nodes, 3 customer classes, and 1500 customers. We then considered a system with more customer classes; due to the greater computational burden, we proposed addressing this problem with simulation. By using a large-scale computing environment (a supercomputer), it was possible to obtain the theoretical solutions for up to three customer classes and verify the simulation accuracy. The parallel simulations' performance-evaluation indices, such as the average number of people in the system, converged to the theoretical values within an acceptable error range after 100,000 simulation hours for systems with four or more customer classes. These results demonstrate that the proposed parallel simulation approach can serve as an accurate and computationally efficient alternative to theoretical solutions for large-scale closed BCMP queueing networks.

## 1 Introduction

Queueing theory has been a fundamental tool for analyzing and optimizing the performance of various systems, such as communication networks, manufacturing systems, and service facilities. The Baskett, Chandy, Muntz, and Palacios (BCMP) queueing network, introduced by Baskett et al. in 1975 [1], is a versatile and widely-used model that extends the earlier works of Jackson [2] and Gordon/Newell [3]. The BCMP network allows for open, closed, and mixed network types, four service disciplines, and multiple customer classes, making it applicable to a wide range of real-world problems [4–7].

([https://github.com/smzn/Limitations-of-Calculating-Theoretical-Solutions-for-Closed-BCMP-Queueing-Networks-and-Verification-](https://github.com/smzn/Limitations-of-Calculating-Theoretical-Solutions-for-Closed-BCMP-Queueing-Networks-and-Verification-)).

**Funding:** This work was supported by JSPS (Japan Society for the Promotion of Science) KAKENHI (grant number JP21K11774). The funders had no role in study design, data collection and analysis, decision to publish, or preparation of the manuscript.

**Competing interests:** The authors have declared that no competing interests exist.

Despite its theoretical elegance, the application of queueing theory to large-scale systems has been hindered by the computational complexity involved. Papadimitriou and Tsitsiklis [8] showed that finding the optimal control policy in a multi-class closed queueing network is an intractable problem with an exponentially large computational time. Consequently, the application of queueing theory to large-scale systems has been limited in practice.

Recently, "waiting"—an essential concept of queuing theory—has been revisited because of its connection to crowding. The global spread of the novel coronavirus has increased awareness of social distance [9]. The importance of managing queues and reducing waiting times has become more evident in the context of the pandemic, as crowding in public spaces can increase the risk of disease transmission [10]. Before the pandemic, crowding while "waiting in line" was accepted as a natural occurrence. However, reduction of crowding is now essential to prevent the spreading of infection. Effective queue management strategies, informed by queueing theory, can play a crucial role in redesigning public spaces and facilities, such as theme parks, shopping malls, and schools, to ensure public safety and well-being [11].

Nearly 50 years after the introduction of the BCMP queueing network, advances in computing power and parallel processing have opened up new possibilities for applying queueing theory to large-scale systems [12, 13]. This study aims to investigate the computational limits of solving large-scale closed BCMP queueing networks using exact methods, such as mean value analysis (MVA) [14], in a modern parallel computing environment. Furthermore, we propose an alternative approach using parallel simulations to approximate performance measures when the number of customer classes is large, making exact solutions computationally intractable [15–17]. The calculation time and memory used for parallel computing were evaluated for various numbers of nodes, customer classes, and people in the network, indicating the required computing resources for different model scales. Calculations involving four or more customer classes are challenging for existing methods; thus, we proposed using simulations for performance evaluation. The simulation results showed sufficiently acceptable errors compared to the theoretical values obtained. Additionally, the simulation approach can be applied when the number of customer classes is four or more, significantly reducing the computational resources required. This alternative simulation method confirms that theoretical values can be computed with significantly fewer computational resources.

Previous simulation studies have proposed the maximum and the minimum number of customers at each node, temporal correlation of the number of customers in the system at each node, and distribution of the number of people when travel time is considered [18]. Combining these with the proposed simulations is expected to provide insights that cannot be obtained with theoretical values, thus making it easier to apply the model to real-world situations.

The rest of this paper is organized as follows. Section 1.1 reviews the related work on computational methods for closed BCMP queueing networks and their applications, while Section 1.2 outlines the contributions of this study. Section 2 describes the methodology, including the definition of a closed BCMP queueing network, the mean value analysis method, the proposed algorithm for parallel computing using MPI, and the parallel simulation approach. Section 3 presents the experimental results and discussion, covering the massively parallel computing environment for MVA, factors affecting the calculation of theoretical solutions, considerations for the number of parallels, and the calculation of performance metrics using parallel simulation. This section also includes an accuracy verification of parallel simulations and explores the use of parallel simulation as an alternative to theoretical values. Additionally, it discusses the limitations of calculating theoretical values for closed BCMP and the effects of parallel simulation. Finally, Section 4 concludes the paper, summarizing the key findings and suggesting potential future research directions.

## 1.1 Related work

Although there are both open and closed types of queueing networks, performance metrics, such as the mean number of people in the system, can be obtained by finding the normalization constant based on the results of the product-form solution [4–6]. For open-type queuing networks, the normalization constant can be easily obtained numerically, and it is possible to make the network large. By contrast, when obtaining the normalization constant for closed-type queuing networks, all combinations that depend on the number of nodes, customer classes, and people in the network must be calculated, which requires massive computational resources [7]. This makes it difficult to upscale the queueing network in a closed system.

Recent studies have focused on developing efficient computational methods for analyzing large-scale closed queueing networks and their applications in various domains. Patel and Bhatha-wala [19] proposed a performance analysis method for closed queuing networks with multiple customer classes and non-exponential service times, extending the applicability of the BCMP model. Li and Fang [20] developed a parallel simulation approach for large-scale closed queuing networks with multi-class customers, demonstrating the scalability of simulation-based methods.

The integration of machine learning techniques with queueing theory has also gained attention in recent years. Osakwe et al. [21] proposed a deep reinforcement learning approach for performance optimization of queueing systems, showcasing the potential of data-driven methods in queueing network control. Sato and Yokota [22] applied closed queueing network models to analyze the performance of smart factories, highlighting the relevance of queueing theory in modern manufacturing systems.

Approximation methods have been developed to tackle the computational complexity of closed BCMP queueing networks. Chen and Yuan [23] proposed an approximate analysis method for closed queueing networks with load-dependent service times, providing a computationally efficient alternative to exact solutions.

Previous studies of computational methods for closed queueing networks, shown in Table 1, have presented numerical calculations on small networks. Moreover, the type of computational environment required was not clearly indicated. In the real world, closed queueing networks tend to be large. Therefore, estimating the computing resources required in the current computing environment is essential.

Typical algorithms for computing closed BCMP are convolution [30] and mean value analysis [31–33]. The convolution method directly calculates performance evaluation indices, such as the stationary distribution and the mean number of people in the system, by finding the normalization constant. However, recursive calculations are often used because of the formula structure for obtaining normalization constants. Although recursive calculations are effective in programming techniques such as memorization [34], they are not effective in parallelizing a large-scale computing environment. The calculation method depends on CPU performance, making it difficult to increase the scale of the model. This is exacerbated by several factorial and power calculations in the formulas and the fact that there is a dropout of orders of magnitude.

By contrast, although the mean value analysis method uses an enormous amount of memory, its calculation structure is suitable for parallel computation, suggesting great potential for large-scale applications. Many approximation algorithms have also been proposed [24–27], some of them using Brownian models [35, 36] or adding complex service conditions [37]. However, in today's large-scale computing environment, sufficient memory and large-scale calculations can be expected without using approximations.

Simulations of closed BCMP have been studied for a long time, largely because of the difficulty of obtaining theoretical values for closed queueing networks, which have been used as an

**Table 1. Network size and computational environment of previous studies in closed queueing networks.**

| Reference number | Publication year | Number of nodes | Number of customer classes | Number of people in network | Algorithm | Computation time | Computing environment |
|---|---|---|---|---|---|---|---|
| [5] | 1990 | 3 | 3 | 20 | Exact and Approximate Closed Multiclass MVA | Exact: 25.93(s) Approximate: 2.03(s) | IBM compatible PC with a 33 MHz Intel 80386 |
| [24] | 2000 | 50 | 2 | 200 | Improved Approximate Mean Value Analysis Library (IAMVAL) | NA | NA |
| [7] | 2003 | 2 | 2 | 4 | Convolution | NA | NA |
| [4] | 2006 | 6 | 3 | 7 | Convolution, MVA, FES, some approximation methods | NA | NA |
| [25] | 2007 | 3 | 4 | 10 | Schweitzer-Bard (S-B) approximation for MVA | NA | NA |
| [26] | 2008 | 30 | 4 | 120 | General Form Linearizer (GFL) algorithms | Presenting some cases | Sun Sparc 20 running SunOS 5.5.1 |
| [27] | 2008 | 10 | 4 | 100 | Conditional Mean Value Analysis (CMVA) algorithm | NA | NA |
| [6] | 2009 | 5 | 1 | 10 | Mean Value Analysis and Convolution Algorithm | NA | NA |
| [28] | 2016 | 4 | 2 | 3 | Mean Value Analysis and Convolution Algorithm | NA | NA |
| [29] | 2019 | 4 | (3, 5) | (15, 6) | One or Two-Phase Class Aggregation | NA | NA |

alternative method [33, 38, 39]. Simulations provide dynamic information that cannot be obtained under static conditions [40, 41], which is crucial when dealing with real-world models. Parallelization has been proposed to speed up simulation [42, 43]; however, the scale of the queueing network has remained small, and in terms of computational resources, the number of parallels has remained limited. Therefore, simulations have not reached a level appropriate for real-world applications.

Several studies have proposed real-world applications of queuing [44], e.g. to urban transportation networks [45] and theme parks [46]. For instance, Mizuno et al. [18, 46] computed a closed queueing network for mobile vehicles in a theme park and performed optimal node placement of mobile vehicles. Nevertheless, only one type of mobile vehicle was used because of computational complexity. The application of machine learning to social systems is also progressing and being increasingly linked with queueing theory. Links with supervised learning (e.g., neural networks [47, 48]) and with the game theory that incorporates customers' experiential behavior [49] have been proposed, but a computational environment for large-scale implementation is essential for social implementation.

The growing use of automation, seen in self-driving cars and in robots performing picking tasks in factories, makes it important to evaluate large, closed queueing networks. If the performance evaluation values for closed BCMP queues can be rapidly calculated, evaluation and optimization of the performance of realistic queueing networks would become feasible.

## 1.2 Contributions of this study

This study makes the following contributions:

i. For closed BCMP, we used MVA, an exact method for obtaining theoretical solutions, to show the limits computable in a modern computing environment. The performance-evaluation index of the closed BCMP could be obtained adequately for large-scale networks,

especially when the number of customer classes was three or less. The scale of the systems considered in this study, in terms of the number of nodes, customer classes, and people in the network, is significantly larger than those in previous studies, as shown in Table 1. This demonstrates the novelty and importance of our work in tackling the challenges of large-scale system analysis.

ii. When the number of customer classes is four or more, the number of combinations in MVA becomes enormous, making it difficult to obtain a theoretical solution. Therefore, simulations were conducted and their results were compared with the theoretical solution.

iii. Using simulations with verified accuracy, we calculated performance metrics for closed BCMP with four or more customer classes. This simulation-based approach serves as an efficient alternative to exact methods for analyzing large-scale closed BCMP queueing networks that were previously intractable.

iv. The degree of simulation convergence was verified using the set simulation error-index to confirm the convergence of the simulation.

v. Using the results of this study, it is possible to choose parameter settings such as the number of nodes, computational resources, and computation time for the construction of a closed BCMP optimization model.

## 2. Materials and methods

This section defines the proposed closed BCMP queueing network and presents a parallel computation algorithm for this large-scale network. We also propose using simulations as an alternative method for calculating the theoretical values.

### 2.1 Definition of a closed BCMP queueing network

The basic model of the closed BCMP queueing network proposed in this study is defined as follows [1]:

i. There are $C$ types of customer classes served in a network, and a customer belongs to one of the classes. There shall be no change of customer class during the process.

ii. There are $N$ nodes in the network.

iii. When the total number of customers in the network is $K$, the number of customers of class $c$ ($1 \leq c \leq C$) in node $n$ ($1 \leq n \leq N$) is $k_{nc} \geq 0$, and $K = \sum_{c=1}^{C} \sum_{n=1}^{N} k_{nc}$. Moreover, $k_n = \sum_{c=1}^{C} k_{nc}, \ k_c = \sum_{n=1}^{N} k_{nc}$.

iv. There shall be no arrival of guests from outside the network, as it is of closed type.

v. Each node consists of a single server performing a first-come-first-service (FCFS).

vi. At location $n$, the service time follows an exponential distribution with service rate $\mu_n$ ($1 \leq n \leq N$), independent of the customer class.

vii. The total arrival rate of class $c$ customers arriving from within the network at node $n$ is $\alpha_{nc}$.

viii. A customer of class $v$ served at node $i$ moves to node $j$ in class $w$ according to a Markov chain $\boldsymbol{R} = (r_{iv,jw})$ that satisfies

$$1 \leq i \leq N, 1 \leq v \leq C, r_{iv,jw} \geq 0, \sum_{j=1}^{N} \sum_{w=1}^{C} r_{iv,jw} = 1. \tag{1}$$

As the class transition of customers is not considered this time, it is assumed that $r_{iv,jw} = 0$ ($v \neq w$).

ix. The arrival rate $\alpha_{iv}$ satisfies the following traffic equation for this closed network:

$$\alpha_{iv} = \sum_{c=1}^{C} \left( \sum_{n=1}^{N} \alpha_{nc} r_{nc,iv} \right), \ (1 \leq i \leq N, \ 1 \leq v \leq C). \tag{2}$$

The state of node $n$ is represented by $\boldsymbol{s_n} = (s_{n1}, s_{n2}, \cdots, s_{nk_n})$. Let $s_{nj}$ denote the class of the $j$th customer in line in order of arrival at node $n$. Let $\boldsymbol{s} = (\boldsymbol{s_1}, \boldsymbol{s_2}, \cdots, \boldsymbol{s_N})$ denote the state in the network and $\boldsymbol{k} = (k_1, k_2, \cdots, k_c)$ denote the number of people by customer class.

From this, the stationary distribution $\pi(\boldsymbol{s})$ in the closed BCMP model is given by

$$\pi(\boldsymbol{s}) = \frac{1}{G(\boldsymbol{k})} \prod_{n=1}^{N} f_n(\boldsymbol{s_n}), \tag{3}$$

where $G(k)$ is the normalization constant

$$G(\boldsymbol{k}) = \sum_{\sum_{n=1}^{N} s_n = k} \prod_{n=1}^{N} f_n(\boldsymbol{s_n}). \tag{4}$$

Additionally, $f_n(\boldsymbol{s_n})$ obeys

$$f_n(\boldsymbol{s_n}) = \prod_{j=1}^{k_n} \frac{\alpha_{ns_{nj}}}{\mu_n}. \tag{5}$$

Eqs (1)–(5) describe the fundamental properties of the closed BCMP queueing network model, as defined in the original paper by Baskett et al. [1].

## 2.2 Calculation of BCMP performance-evaluation index using mean value analysis

The mean number of people in the system for customer class $c$ at node $n$ using the mean value analysis method is calculated using the following updated formulas [4]: The mean intra-system time $T_{n,c}(\boldsymbol{k})$ ($1 \leq n \leq N, 1 \leq c \leq C$) is

$$T_{n,c}(\boldsymbol{k}) = \frac{1}{\mu_n} \left( 1 + \sum_{v=1}^{C} L_{n,v}(\boldsymbol{k} - 1_v) \right). \tag{6}$$

The throughput $\lambda_c(\boldsymbol{k})$ is

$$\lambda_c(\boldsymbol{k}) = \frac{k_c}{\sum_{i=1}^{N} \alpha_{ic} T_{i,c}(\boldsymbol{k})}. \tag{7}$$

The mean number of people in the system $L_{n,c}(\boldsymbol{k})$ is

$$L_{n,c}(\boldsymbol{k}) = \lambda_c(\boldsymbol{k}) T_{n,c}(\boldsymbol{k}) \alpha_{nc}. \tag{8}$$

Eqs (6)–(8) represent the basic steps of the mean value analysis (MVA) method for closed BCMP queueing networks, as described in Bolch et al. [4].

## 2.3 Proposed algorithm for using parallel computing (MPI) in MVA

The mean value analysis method can calculate performance evaluation indicators, such as the mean number of people in the system, through a simple calculation process. However, as the number of sites, classes, and people in the system increases and the scale of the model grows, the computation time becomes longer. Therefore, parallel computation is used to reduce computation time. The following algorithm describes the parallel computation flow when Message Passing Interface (MPI) [50] is applied to MVA:

A. Registration of the configuration information

 a. Specify the number of nodes $N$; the number of classes $C$; the number of customers by class $k_c$ ($1 \leq c \leq C$); the node service rate $\mu_n$ ($1 \leq n \leq N$); and the number of parallels $MP$. The process number is $mp$ ($0 \leq mp \leq MP-1$), where $mp = 0$ is called the root process and is represented by $mp_0$.

 b. $mp_0$: Set transition probability $R$, find arrival rate $\alpha_{nc}$, and broadcast to other processes.

B. MVA calculation *for j in 1,···, K*

 a. For each iteration, $mp_0$ obtains a set satisfying
$index = j, (1 \leq j \leq K), k_c \leq j, (1 \leq c \leq C)$. For this, we define the set
$K_c = \{0, 1, 2, \cdots, k_c\}, (1 \leq c \leq C)$. Using this, the direct product set $K^{(j)}$ of the number of classes $C$ of $K_c$ follows

$$K^{(j)} = \{(k_1^{(j)}, k_2^{(j)}, \cdots, k_C^{(j)}) | K^{(j)} \in K_1 \times \cdots \times K_C, k_c^{(j)} \in K_c, \sum\nolimits_{c=1}^{C} k_c^{(j)} = j, 1 \leq c \leq C\}, (9)$$

when the number of people in the system is $j$. Thus, $k_m^{(j)} \in K^{(j)}, 1 \leq m \leq |K^{(j)}|$, is obtained.

 b. $mp_0$ divides $K^{(j)}$ into $K_i^{(j)}, (0 \leq i \leq MP - 1)$, and passes it to each process. $k_i^{(j)}(m) = \{(k_1^{(j)}(m), k_2^{(j)}(m), \cdots, k_C^{(j)}(m)) | k_i^{(j)}(m) \in K_i^{(j)}, 1 \leq m \leq |K_i^{(j)}|\}$,
where $k_i^{(j)}(m)$ is each $m$-th class people vector passed to process $i$ when the total number of customers is $j$.

 c. Using $k_i^{(j)}(m)$ passed in each process, find $L_{n,c}(k_i^{(j)}(m)), (n = 1 \leq n \leq N, 1 \leq c \leq C)$ from Eqs (6)–(8).

 d. $mp_0$ aggregates the $L_{n,c}(k_i^{(j)}(m))$ obtained by each process and broadcasts $L_{n,c}(\cdot)$ to each process.

 e. If *index* = $K$ is exceeded, the iteration is terminated, and performance evaluation indicators, such as the average number of people in the system, are stored.

Fig 1 illustrates the detailed steps of the parallel computation algorithm for MVA using MPI, providing a visual representation of the process described in this section. Eq (9) is an original set definition introduced in this study to calculate the overlap combination for each iteration $j$ in the proposed parallel computation algorithm for MVA. In the above algorithm, Eq (9) in step B.a calculates the overlap combination for each iteration $j$ such that the number of customers in each class does not exceed $j$, and the total number of customers in all classes is $j$. In B.b, the overlap combination obtained in B.a is distributed to parallel process $i$. In B.c, the calculation of Eqs (6)–(8) is performed in each parallel process. Finally, in B.d, the iterative calculation is performed using the root process $mp_0$ to aggregate.

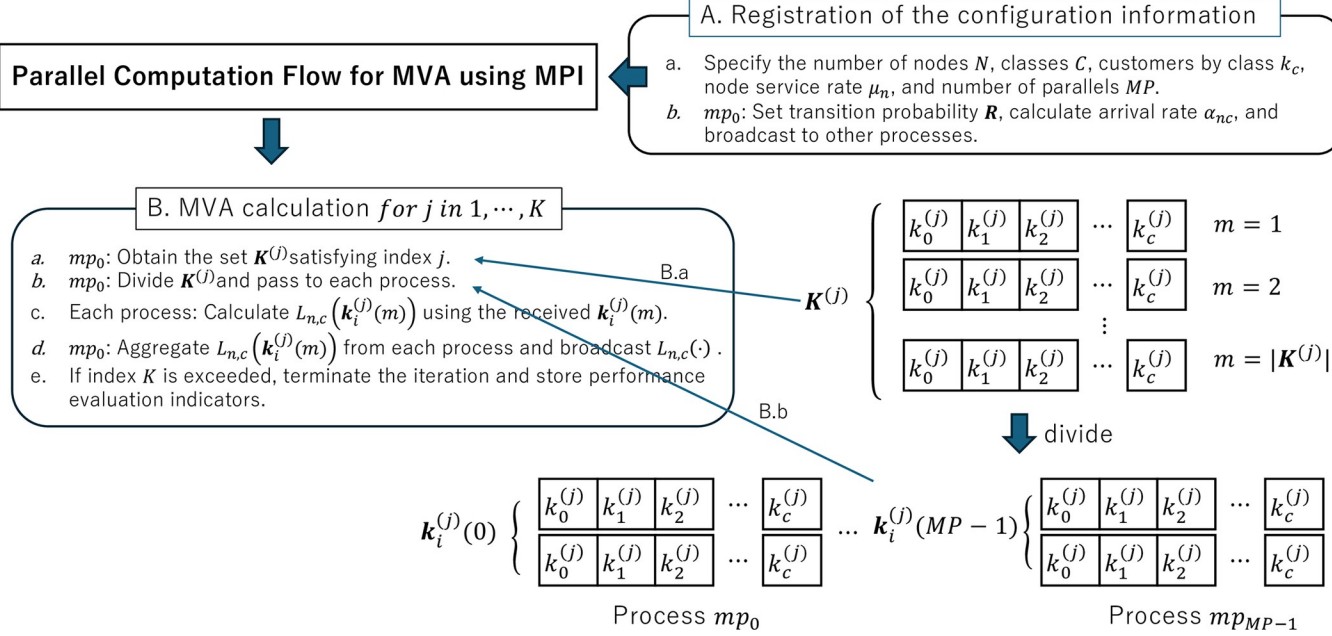

**Fig 1. Flowchart of the parallel computation algorithm for Mean Value Analysis (MVA) using Message Passing Interface (MPI).**

## 2.4 Calculation of performance metrics using parallel simulation

When the network becomes large, e.g., when there is an increase in the number of customer classes, the computing resources required become enormous, making computation challenging. Therefore, as an alternative to the theoretical calculation method of MVA, we propose using the mean number of people in the system in parallel simulations when calculating performance evaluation indices. The process is as follows:

A. Registration of the configuration information

 a. Specify the number of nodes $N$; the number of classes $C$; the number of customers by class $k_c$ ($1 \leq c \leq C$); the node service rate $\mu_n$ ($1 \leq n \leq N$); the number of simulations $S$; and the simulation end condition $\varepsilon$. The process number is $mp$ ($0 \leq mp \leq MP-1$), where $mp = 0$ is called the root process and is represented by $mp_0$.

 b. $mp_0$: Set transition probability $\mathbf{R}$, find arrival rate $\alpha_{nc}$, and broadcast to other processes.

B. Perform event-driven parallel simulation

 a. Initial event setup: In each process, randomly assign customers in the network to nodes and give the first customer at each node a randomly distributed service time.

 b. Event processing: In each process, the minimum-service-time customer is served from the entire node; after the service is completed, the customer is moved to another node and the simulation time is updated.

 i. Calculate performance indicators such as the mean number of people in the system for each process.

 ii. In $mp_0$, $L_{ir}^{(s)}$ is used to calculate the mean $\bar{L}_{n,c}$ ($n = 1, \cdots, N, c = 1, \cdots, C$), where $L_{ir}^{(s)}$ is the mean number of people in the system for node $n$ and class $c$ of simulation number $s$.

iii. The allowable error in the number of people is given by

$$\frac{1}{N \cdot C \cdot S} \sqrt{\sum_{n=1}^{N} \sum_{c=1}^{C} \sum_{s=1}^{S} \left(L_{n,c}^{(s)} - \bar{L}_{n,c}\right)^2} < \varepsilon. \tag{10}$$

Repeat until Eq (10) is satisfied.

Eq (10) is an original formula for evaluating the error precision of the parallel simulation approach proposed in this study. As shown in Fig 2, the algorithm consists of two main parts: (A) Registration of the configuration information, and (B) Performing the event-driven parallel simulation. The flowchart clearly depicts the iterative nature of the simulation process and the error checking mechanism, which ensures the accuracy of the results. This visual representation enhances the understanding of our proposed parallel simulation approach for BCMP queueing networks.

## 3. Results and discussion

In this chapter, we present and discuss the results of our study on the limitations of calculating theoretical values for closed BCMP queueing networks and the effectiveness of parallel simulation as an alternative method. We begin by describing the massively parallel computing

### A. Registration of the configuration information

a. Specify the number of nodes $N$, classes $C$, customers by class $k_c$, node service rate $\mu_n$, and number of parallels $MP$.

b. $mp_0$: Set transition probability $\boldsymbol{R}$, calculate arrival rate $\alpha_{nc}$, and broadcast to other processes.

### Event-Driven Parallel Simulation for BCMP

### B. Perform event-driven parallel simulation

a. Initial Event Setup: Randomly assign customers and set service times.

b. Event Processing: Serve minimum − service −
time customer, move to another node, update time.

 i. Calculate indicators for each process.

 ii. $mp_0$: Calculate mean $\bar{L}_{n,c}$ using $L_{ir}^{(s)}$ . 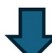

 iii. Check error: $\frac{1}{N \cdot C \cdot S} \sqrt{\sum_{n=1}^{N} \sum_{c=1}^{C} \sum_{s=1}^{S} (L_{n,c}^{(s)} - \bar{L}_{n,c})^2} < \varepsilon$ 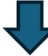
Repeat if needed.

**Fig 2. Flowchart of the event-driven parallel simulation algorithm for BCMP queueing networks.**

environment used for the mean value analysis (MVA) in Section 3.1. In Section 3.2, we examine the factors affecting the calculation of the theoretical solution for a closed BCMP queueing network, followed by a consideration of the number of parallels for MVA parallel computation in Section 3.3.

Next, we propose the use of parallel simulation to calculate performance metrics for closed BCMP when the number of customer classes is large, making MVA computationally challenging. We verify the accuracy of parallel simulations in Section 3.4.1 and present the results of using parallel simulation as an alternative to theoretical values in Section 3.4.2.

Finally, in Section 3.5, we discuss the limitations of calculating theoretical values for closed BCMP and the effects of parallel simulation, highlighting the contributions and implications of our study.

### 3.1 Massively parallel computing environment for MVA

In this study, the large-scale parallel computing environment for MVA was OCTOPUS [51], a large-scale computing system at the Cybermedia Center of Osaka University. The specifications of the computing environment are listed in Table 2.

### 3.2 Factors affecting the calculation of the theoretical solution for a closed BCMP queueing network

Here, we present the results of applying MPI to MVA to compute closed BCMP theoretical solutions. Table 3 summarizes the computation time, memory used, and number of combinations when $N$, $C$, and $K$ are varied for 128 parallels. For example, consider the case $N = 33$, $C = 3$, $K = 500$, and $MP = 128$; the number of people per customer class was assumed to be equally $K/C$ for every class. In this case, the computation time in MVA was 4166.07 seconds (approximately 70 minutes), and the maximum memory used was 114.44 GB.

Fig 3 shows the change in computation time as $N$ and $K$ increased. Note that the computation time grew proportionally to the increase in $N$ (Fig 4). The computational complexity $O(2C(N-1)\prod_{c=1}^{C}(k_c+1))$ [4] also increased linearly as $O_N = D_1 N$, ($D_1$ is constant). However, Fig 5 shows that the computation time increased exponentially with $K$. In this case, the computational complexity was $O_K = D_2(D_3+K)^C$, ($D_2$ and $D_3$ are constants). The calculated amount of memory used was $O(N \cdot \prod_{c=1}^{C}(k_c+1))$ [4]. Thus, the calculation results show that the increase in calculation time is proportional to $N$ but grows exponentially when $K$ is increased.

**Table 2.** *Massively parallel computing environment for MVA.*

| Item | Description |
|---|---|
| Programming language | Python 3.6.13 |
| Library used | mpi4py 3.1.3 |
| Computing environment | OCTOPUS (Osaka University) |
| | Nodes with large-capacity main memory |
| | 2 nodes (16.38 TFLOPS) |
| Processor information | Intel Xeon Platinum 8153 |
| | (Skylake / 2.0 GHz 16 Core) 8 units |
| Memory | 6 TB/Node |
| Number of nodes used | 2 (max) |
| Number of cores used | 1. parallels per node |

**Table 3.** *Results with 128 parallels and various values of* N, C, *and* K.

| N | C | K | Computation time (s) | Maximum memory used (GB) | Number of combinations (max, total) |
|---|---|---|---|---|---|
| 33 | 1 | 500 | 514.28 | 48.95 | 1, 501 |
| 33 | 2 | 500 | 546.18 | 49.57 | 251, 63001 |
| 33 | 3 | 500 | 4166.07 | 114.44 | 21084、4713408 |
| 66 | 3 | 500 | 7413.16 | 193.29 | 21084、4713408 |
| 100 | 3 | 500 | 10292.22 | 266.64 | 21084、4713408 |
| 133 | 3 | 500 | 13354.99 | 313.77 | 21084、4713408 |
| 166 | 3 | 500 | 16041.46 | 363.84 | 21084、4713408 |
| 200 | 3 | 500 | 19144.92 | 456.63 | 21084、4713408 |
| 233 | 3 | 500 | 22191.55 | 505.93 | 21084、4713408 |
| 266 | 3 | 500 | 25071.72 | 553.32 | 21084、4713408 |
| 300 | 3 | 500 | 28388.36 | 603.48 | 21084、4713408 |
| 33 | 3 | 250 | 667.99 | 84.74 | 5334、599759 |
| 33 | 3 | 750 | 15182.38 | 205.88 | 47251、15813250 |
| 33 | 3 | 1000 | 37857.74 | 313.45 | 83834、37371259 |
| 33 | 3 | 1250 | 79495.01 | 470.02 | 130834、72859907 |
| 33 | 3 | 1500 | 146798.86 | 604.51 | 188251、125751500 |

Table 3 shows the results up to $C = 3$. (The calculation could not be performed for $C \geq 4$ in this calculation environment.) With $N = 33$ and $K = 500$, the total number of combinations when calculating with MVA was 501 for $C = 1$, 63,001 for $C = 2$, 4,713,408 for $C = 3$, and 252,047,375 for $C = 4$, requiring several combination calculations. The number of combinations and the memory that can be held by $C \geq 4$ must be allocated, which is a significant burden for a computing environment.

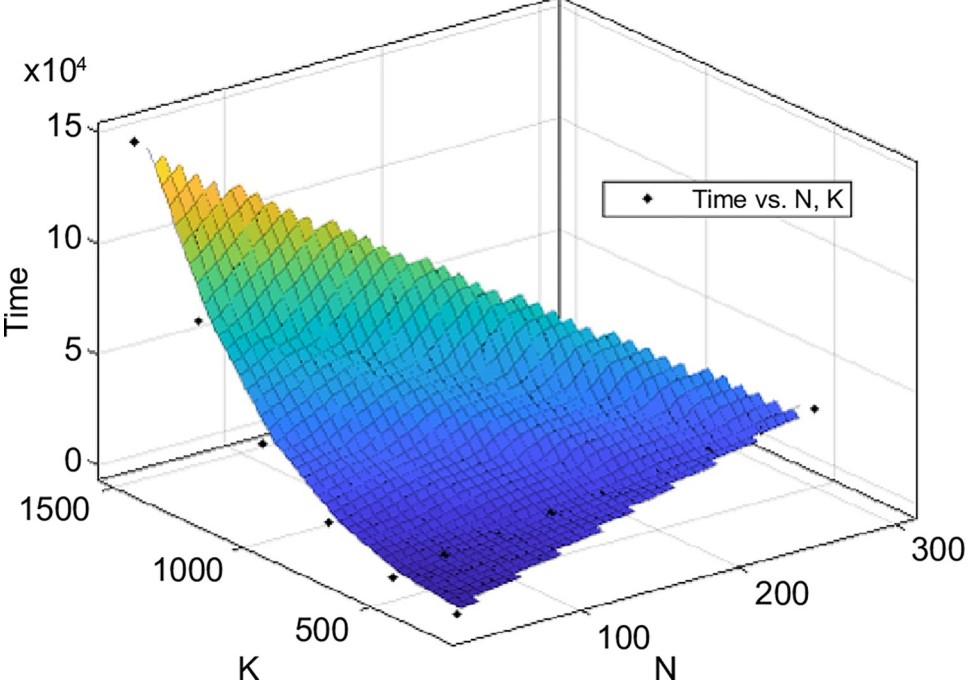

**Fig 3. Change in computation time for an increase in number of nodes** $N$ **and number of people in system** $K$ **($C = 3$).**

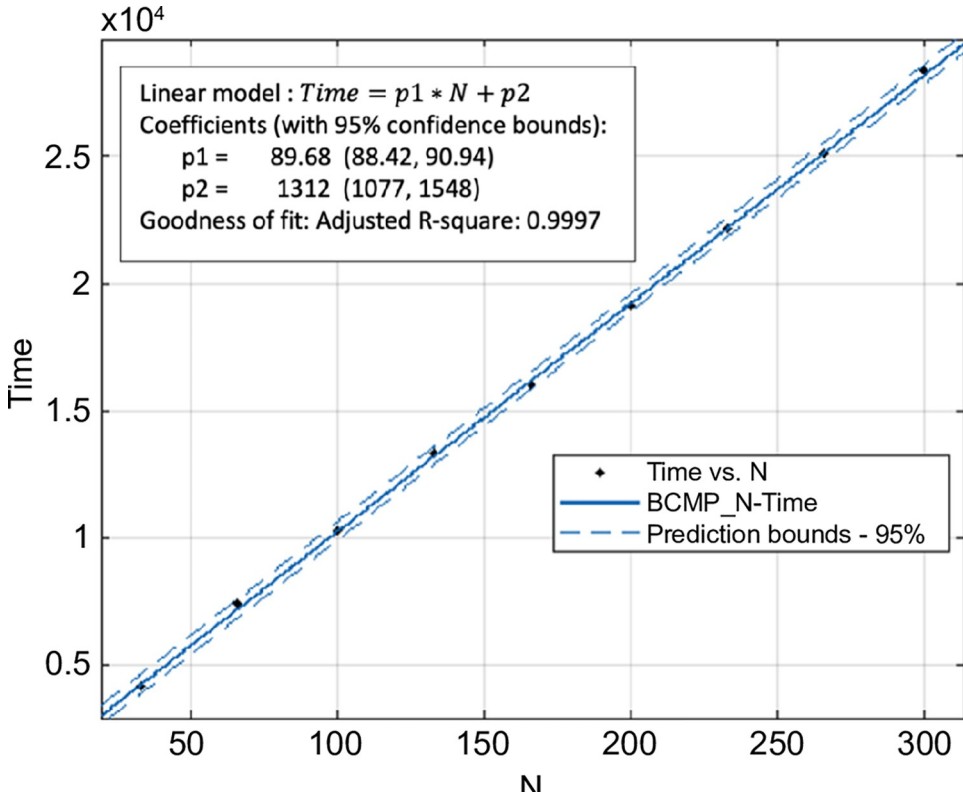

**Fig 4. Variation of computation time with respect to $N$ ($K = 500, C = 3$).**

### 3.3 Consideration of the number of parallels for MVA parallel computation for closed BCMP

The effect on computational time of the number of parallels in the parallelization of MVA for the closed BCMP is illustrated in Table 4. In particular, the table summarizes the computation time for the cases $N = 33, C = 3$, and $K = 500$, varying the number of parallels from 1 to 256. Because the maximum number of parallels on a single compute server is 128, two compute servers were used when the number of parallels was larger than this. Increasing the number of parallels reduces the amount of computation time required. However, the amount of memory used increases, and consequently, so does the utilization cost. Considering the rate of decrease in computation time, memory usage, and computer usage fees, 96~128 parallels were considered appropriate for using one compute server.

### 3.4 Calculation of performance metrics for closed BCMP using parallel simulation

In the closed BCMP, the calculation of performance evaluation indices using parallel computation in MVA has become a major burden due to the increase in computation volume as the number of customer classes increases. Herein, instead of calculating the performance evaluation index strictly using MVA, we consider calculating it using parallel simulation. First, the simulation accuracy is confirmed using the obtained MVA calculation results. Next, performance evaluation indexes are calculated for patterns with a large number of customer classes that could not be obtained using MVA. The simulation can be performed even without the

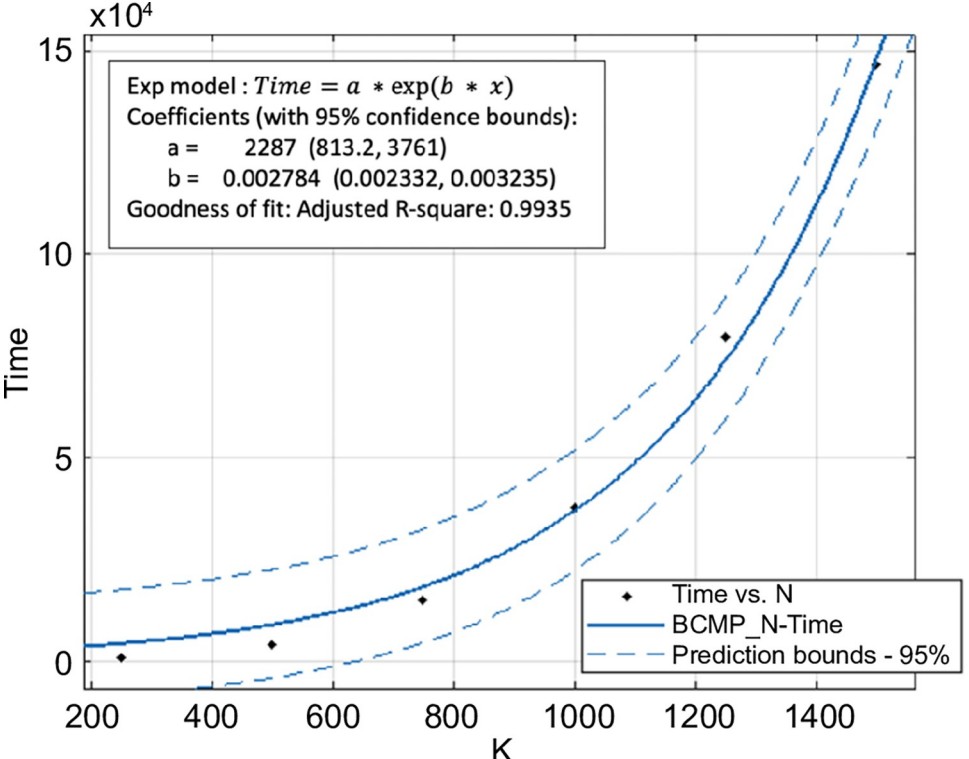

**Fig 5. Variation of computation time with respect to $K$ ($N = 33, C = 3$).**

large-scale computing environment used in the parallel computation of MVA, although the accuracy must be carefully verified. The following simulation results were calculated on a Mac Pro (model: Late 2013, processor: 3GHz 8-core Intel Xeon E5, memory: 32GB 1866MHz DDR3), with eight parallel calculations.

**3.4.1 Accuracy verification of parallel simulations.**  First, simulations with $N = 33, C = 3$, and $K = 500$ were performed to calculate the error relative to the theoretical values calculated by MVA. With a total number of people $K$ in the network, the error tolerance $E_p$ at each node and each class is

$$E_p = \frac{K}{N \cdot C} \cdot p. \tag{11}$$

If it is assumed that $p = 0.3$, an error of 30.0% is assigned to each node and class on average. Subsequently, for $K = 500$, Eq (11) for each node and class yields 4.54 for $C = 1$; 2.27 for $C = 2$; and 1.51 for $C = 3$. The root mean squared error between the simulation and theoretical values

**Table 4. Computation time with different number of parallels ($N = 33, C = 3, K = 500$).**

| Number of parallels | 1 | 32 | 64 | 96 | 128 | 160 | 192 | 224 | 256 |
|---|---|---|---|---|---|---|---|---|---|
| Number of compute servers used | 1 | 1 | 1 | 1 | 1 | 2 | 2 | 2 | 2 |
| Number of parallels / server | 1 | 32 | 64 | 96 | 128 | 80 | 96 | 112 | 128 |
| Maximum memory used (GB) | 43.44 | 64.81 | 87.08 | 109.52 | 114.44 | 132.96 | 133.55 | 134.24 | 134.64 |
| Computation time (s) | 49483.31 | 5290.10 | 4428.09 | 4129.25 | 4166.07 | 4119.36 | 4032.26 | 3986.72 | 3967.71 |

**Table 5. Simulation accuracy for varying $C$ when $N = 33, K = 500$.**

| | Simulation time | 1000 | 5000 | 10000 | 20000 | 30000 | 40000 | 50000 | 100000 | $E_p$ achievement time |
|---|---|---|---|---|---|---|---|---|---|---|
| (12) | $C = 1$ $E_p = 4.54$ | 45.446 | 36.344 | 25.460 | 13.883 | 8.934 | 7.202 | 5.354 | **3.346** | 69200 |
| | $C = 2$ $E_p = 2.27$ | 27.460 | 18.532 | 11.457 | 5.895 | 4.124 | 2.875 | **2.215** | **1.043** | 48700 |
| | $C = 3$ $E_p = 1.51$ | 13.333 | 9.359 | 5.866 | 2.970 | 2.018 | 1.737 | **1.452** | **0.640** | 48350 |

for each node and class is

$$\frac{1}{N \cdot C} \sqrt{\sum_{n=1}^{N} \sum_{c=1}^{C} \left( \bar{L}_{n,c} - L_{n,c}^{T} \right)^2}, \tag{12}$$

where $L_{n,c}^{T}$ represents the theoretical value for node $n$ and class $c$. Each simulation was performed eight times, and the mean $\bar{L}_{n,c}$ over the six simulations, excluding the best and worst cases, was used. Table 5 lists the result obtained using Eq (12). The bolded area is the value of Eq (12) below $E_p$. Approximately 7,000 simulation hours were required to reach the error tolerance. Similarly, the simulation accuracy was verified when $N$ and $K$ increased. Table 6 summarizes the change in the value of Eq (12) when $N$ and $K$ were varied for $C = 3$. In all cases, the simulations were found to be sufficiently accurate.

**3.4.2 Alternative theoretical value using parallel simulation.** Because the simulation accuracy was confirmed in the previous section, simulations were performed for $C \geq 4$, a challenging scenario to compute. As a performance-evaluation index, the mean number of people in the network was calculated. The simulation considered eight simultaneous independent simulations, and the average $\bar{L}_{n,c} (1 \leq n \leq N, 1 \leq c \leq C)$ was obtained from the mean number of people in the system obtained for each simulation value. The squared error with that average was obtained, and the remaining six simulation values, excluding those with the largest and smallest squared errors, were used to calculate the error of the simulation values at each time from Eq (10). Table 7 summarizes the change in Eq (10) when the number of classes increased under $N = 33$ and $K = 500$. In Eq (10), even when $\varepsilon = 0.01$, the value converged after 100,000 simulation hours, and an approximation to the theoretical value was obtained.

Consider the simulation results for $C = 4$, shown in Fig 6. The horizontal axis of the figure is the standard deviation of the average number of customers in the system obtained in the simulation for each process, averaged over all processes. The vertical axis is the value of Eq (12). From this, we can see that larger values along the horizontal axis correspond to smaller root-mean-square errors. The correlation coefficient between the two axes was −0.8920. The transition probability matrix of this simulation was generated for each program. It is easier for a simulation to converge when the queueing network is characterized by locations where the number of customers in the system is concentrated. Thus, this simulation is more likely to

**Table 6. Simulation accuracy for varying $N$ and $K$ when $C = 3$ ($Eq(12)$).**

| | Simulation time | 1000 | 5000 | 10000 | 20000 | 30000 | 40000 | 50000 | 100000 | $E_p$ achievement time |
|---|---|---|---|---|---|---|---|---|---|---|
| $K = 500$ | $N = 66$ $E_p = 0.75$ | 9.099 | 6.274 | 3.787 | 1.845 | 1.212 | 0.962 | 0.874 | **0.476** | 58600 |
| | $N = 100$ $E_p = 0.50$ | 6.063 | 2.372 | 1.425 | 0.683 | 0.538 | **0.476** | **0.353** | **0.131** | 38150 |
| | $N = 133$ $E_p = 0.37$ | 0.979 | 1.043 | 0.731 | 0.495 | **0.301** | **0.296** | **0.255** | **0.158** | 25700 |
| $N = 33$ | $K = 750$ $E_p = 2.27$ | 27.286 | 20.441 | 17.511 | 11.918 | 9.170 | 7.730 | 6.110 | 2.667 | 119950 |
| | $K = 1000$ $E_p = 3.03$ | 35.602 | 32.126 | 28.541 | 22.842 | 17.583 | 13.540 | 10.788 | 4.039 | 119250 |
| | $K = 1250$ $E_p = 3.78$ | 56.628 | 43.676 | 31.999 | 17.764 | 11.791 | 8.854 | 7.054 | **3.453** | 91650 |

**Table 7. The change in the value of Eq ([10]) for the number of classes $C \geq 4$ ($N = 33$, $K = 500$, $\varepsilon = 0.01$).**

| Class\ Simulation time | 1000 | 5000 | 10000 | 20000 | 30000 | 40000 | 50000 | 100000 |
|---|---|---|---|---|---|---|---|---|
| C = 4 | 0.0937 | 0.0993 | 0.0912 | 0.0729 | 0.0791 | 0.0360 | 0.0122 | 0.00231 |
| C = 5 | 0.0614 | 0.0713 | 0.0734 | 0.0751 | 0.0515 | 0.0460 | 0.0390 | 0.00371 |
| C = 6 | 0.0487 | 0.0423 | 0.0379 | 0.0348 | 0.0233 | 0.0195 | 0.0153 | 0.00239 |
| C = 7 | 0.0331 | 0.0418 | 0.0366 | 0.0185 | 0.0275 | 0.0181 | 0.0096 | 0.00467 |
| C = 8 | 0.0284 | 0.0282 | 0.0197 | 0.0197 | 0.0196 | 0.0112 | 0.0132 | 0.00589 |

obtain good values in environments with localized crowding, thus increasing the accuracy of congestion assessment.

The real-time rate is the ratio of the real time taken in the simulation to the simulation time. When $C = 3, K = 500$, and simulation time was 100000, the real time rate was 0.8200 for $N = 33$; 12.1941 for $N = 100$; and 26.7621 for $N = 133$. The impact of $N$ was larger than that of $C$ because $N > C$ and the transition probability, which goes as $o((N \cdot C)^2)$, must be used when a customer moves to the next node after service is completed.

## 3.5 Limitations of calculating theoretical values for closed BCMP and effects of parallel simulation

The computational environment to calculate the theoretical values for the closed BCMP was described in Section 3.1, and the results obtained in that environment were verified in Section 3.2. We also showed the appropriate computational environment for the closed BCMP in Section 3.3. In the presented computational environment, $C = 3$ is the limit for $N = 33$ and $K = 500$. In Section 3.4, we confirmed the accuracy of the simulation using the values obtained in Section 3.2. Moreover, we obtained simulation times for which the simulated values could

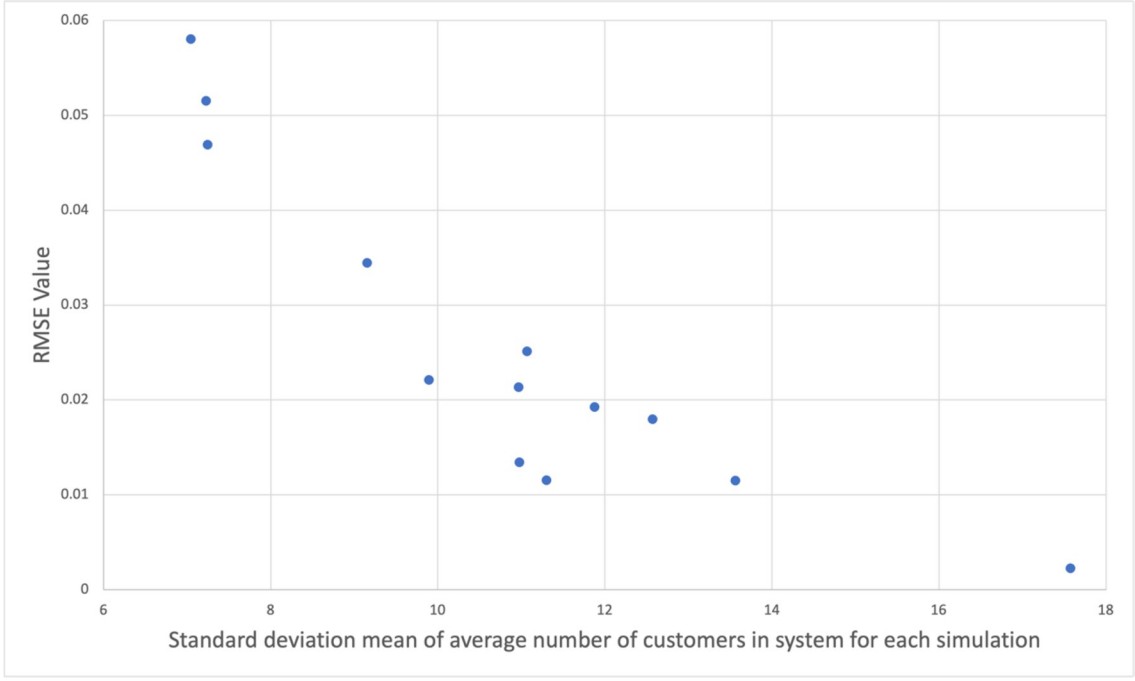

**Fig 6. Root-mean-square error of simulation relative to theoretical value, plotted against the mean of the standard deviation of the number of customers in the mean system for each simulation.**

replace the theoretical values. The results in these sections show the limit of the theoretical value calculation for the closed BCMP using the computational environment. Moreover, we showed that simulation can replace the theoretical value when $C \geq 4$ or more for $N = 33$ and $K = 500$. Furthermore, we have shown that, to obtain the theoretical value of the closed BCMP, an exact method, such as MVA, can be used, rather than an approximate method—even for a larger closed BCMP model than in previous studies. Furthermore, the computational environment and method are explicit and implemented in parallel simulation. This environment can be used as an alternative for models where calculations are challenging or impossible.

In a closed BCMP, simulation requires fewer computational resources and provides dynamic information, but requires longer actual computation time. The closed BCMP model can be utilized in many situations using the method of calculating performance evaluation indices for parallel computation of MVA. This study showcases the limitations of such parallel computation; moreover, the simulation accuracy is verified in this study according to the parameters of the closed BCMP.

## 4. Conclusion

This study performed the parallelization and simulation of MVA to BCMP, the traditionally used and a representative model of a closed queueing network, to scale up the closed BCMP successfully in a large-scale computing environment. We could accommodate changes in parameters, such as the number of nodes and people in the network, up to realistic values. The computation time and memory used for parameter changes were also found, indicating the necessary preparation for the computation environment. However, in this case, the number of customer classes was limited to three because MVA consumes a large amount of memory for larger numbers than this. Therefore, to obtain the mean number of people in a system with more than four customer classes, we performed simulations with the theoretical values of the obtained range and confirmed that the simulation values converged within the acceptable range. This indicates that the simulation method proposed in this study can be used as an alternative to calculating theoretical values of performance-evaluation indicators in a closed BCMP. This has important practical implications: although the number of customer classes is expected to increase as modern society becomes more diverse, performance evaluation indices, such as the mean number of people in the system, can still be obtained using highly accurate simulations.

This paper contributes to the literature in that it reports on the possibility of scaling queueing theory to real-world applications. Queueing theory is mathematically compelling but hard to apply in the real world because of the flexibility of its definition and its large scale. As shown in this paper, simulation allows flexible BCMP to be constructed on a large scale.

Queueing theory is also applicable to optimization, because it can calculate performance evaluation indices such as the mean number of people in the system. The application of queueing theory to optimization has nevertheless been limited, primarily because of its computational complexity. During optimization, the objective function is iteratively calculated several times. If the performance evaluation value of the closed BCMP is used for the objective function and only the computation time of one MVA is considered, the computation time to obtain the optimal solution is substantially larger. Here, we considered a large-scale closed BCMP model; its computation time was reduced by parallel computation. Therefore, this study has shown a computational environment that increases the possibility of applying queuing theory to optimization.

In the present study, the number of servers for each node was limited to one. Increasing the number of servers might provide more realistic values. Using service types other than FCFS

and allowing for customer class transition should be considered for handling realistic models. The use of MVA also makes it difficult to calculate higher-order moments. Higher-order moments in conjunction with the normalization constants used in the convolution method could be calculated to evaluate realistic models. In the computational environment, memory space savings in recursive calculations in MVA and the use of GPUs are required for more generalized, large-scale closed BCMP calculations.

## Acknowledgments

This work was (partly) achieved through the use of OCTOPUS at the Cybermedia Center, Osaka University.

## Author Contributions

**Conceptualization:** Shinya Mizuno.

**Data curation:** Haruka Ohba.

**Formal analysis:** Shinya Mizuno.

**Funding acquisition:** Shinya Mizuno.

**Investigation:** Shinya Mizuno, Haruka Ohba.

**Methodology:** Shinya Mizuno.

**Project administration:** Shinya Mizuno.

**Resources:** Shinya Mizuno.

**Software:** Haruka Ohba.

**Supervision:** Shinya Mizuno.

**Validation:** Shinya Mizuno.

**Visualization:** Haruka Ohba.

**Writing – original draft:** Shinya Mizuno.

**Writing – review & editing:** Shinya Mizuno, Haruka Ohba.

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
