## [Decision Letter · Decision Letter 0]

28 May 2024

PONE-D-24-06423Limitations of Calculating Theoretical Solutions for Closed BCMP Queueing Networks and Verification of Alternative Theoretical Values by Parallel SimulationPLOS ONE

Dear Dr. Mizuno,

Thank you for submitting your manuscript to PLOS ONE. After careful consideration, we feel that it has merit but does not fully meet PLOS ONE’s publication criteria as it currently stands. Therefore, we invite you to submit a revised version of the manuscript that addresses the points raised during the review process.

We look forward to receiving your revised manuscript.

Kind regards,

Palaniyappan Sathyaprakash, Ph.D

Academic Editor

PLOS ONE

Journal Requirements:

3. Thank you for stating the following financial disclosure: "This work was supported by JSPS KAKENHI (grant number JP21K11774). "

4. Please expand the acronym “JSPS” (as indicated in your financial disclosure) so that it states the name of your funders in full.

5. Thank you for stating the following in the Acknowledgments Section of your manuscript: "This work was supported by JSPS KAKENHI (grant number JP21K11774)"

Please remove any funding-related text from the manuscript and let us know how you would like to update your Funding Statement. Currently, your Funding Statement reads as follows: "This work was supported by JSPS KAKENHI (grant number JP21K11774)"

Additional Editor Comments:

To make the "Limitations of Calculating Theoretical Solutions for Closed BCMP Queueing Networks and Verification of Alternative Theoretical Values by Parallel Simulation." balanced, include references from 2019 to 2024

Reviewers' comments:

Reviewer's Responses to Questions

**Comments to the Author**

1. Is the manuscript technically sound, and do the data support the conclusions?

Reviewer #1: Partly

Reviewer #2: Yes

2. Has the statistical analysis been performed appropriately and rigorously? 

Reviewer #1: No

Reviewer #2: I Don't Know

3. Have the authors made all data underlying the findings in their manuscript fully available?

Reviewer #1: Yes

Reviewer #2: Yes

4. Is the manuscript presented in an intelligible fashion and written in standard English?

Reviewer #1: No

Reviewer #2: Yes

5. Review Comments to the Author

Reviewer #1: This paper is applied a closed BCMP queueing network to a real-world model and examining the limitations of the theoretical solution, The paper needs some improvements as listed below.

1. The abstract needs to be a little bit more precise and numerically conclusive.

2. Introduction can be more impressive in terms of Identifying the Research Problem or Gap, State the Research Objectives, Outline the Scope and Methodology, Highlight the Significance of the Study and Provide an Overview of the Paper Structure,

so, author should rewrite the introduction.

3. Related work should be biased on latest work and they should cite the latest research papers too.

4. In this paper, the author provided a theoretical and practical comparison of the results, whether the author should compare their results with the other (same area) paper’s results because in research papers we are working to improve the already done work and result.

5. Overall, an English translation is grammatically sound but the order of placement of topics often goes wrong and reading patterns get out of hand.

6. Since the author did not cite any sources for equations No. 1 through No. 9, it is unclear how real and true they are.

7. Working steps (algorithm) is missing.

8. Include Flow chart and symbol table with this work.

9. Overall, an English translation is grammatically sound but the order of placement of topics often goes wrong and reading patterns get out of hand.

Reviewer #2: This present paper talks about the parallelization and simulation of MVA to BCMP, the traditionally used and a representative model of a closed queueing network. This work is interesting, however it has few lacks that need to be fixed.

- The introduction lacks of references.(from lines 53 to 60)

- The font size in table 1 need to adjusted in such away that the table can be readable easily.

- The ralated work section, is not up to date, please update it. The last reference you are using is since 2019 and there have been few works since then related to the closed BCMP queueing network.

- It would also interesting that the authors add a flow chart of the process of their proposed approach on the system solved used.

- Figures 1, 2 and 3 are not placed correctly in the paper.

- The presented results are promising, however, for a better fainess of the study, the authors are requested to do a comparison study to state of the art with other approaches (if there exist few) for the same datasets in order to show the performance quality of the proposed algorithm.

6. PLOS authors have the option to publish the peer review history of their article (what does this mean?). If published, this will include your full peer review and any attached files.

Reviewer #1: No

Reviewer #2: No

---

## [Author Response · Author response to Decision Letter 0]

9 Jul 2024

Please refer to the attached detailed response letter for a point-by-point response to the reviewers' comments and suggestions. We have carefully addressed all the comments and made the necessary revisions to our manuscript.

Key Revisions:

Abstract: Revised to include more precise and numerically conclusive information.

Introduction: Rewritten to better identify the research problem, state objectives, outline methodology, highlight the significance, and provide an overview of the paper structure.

Related Work: Updated with recent literature (2019-2024) and expanded to cover broader topics including non-exponential service times, parallel simulation approaches, and the impact of COVID-19 on queueing theory applications.

Theoretical and Practical Comparison: Emphasized the unique scale of our study compared to previous works and highlighted the contributions of our parallelized Mean Value Analysis (MVA) algorithm.

Figures and Tables: Adjusted font sizes for readability and repositioned figures to immediately follow their respective references in the text.

Equations: Added citations for all equations to clarify their origin and reliability.

Algorithm Steps and Flowcharts: Included detailed steps of the proposed algorithms and added flowcharts to enhance clarity and reproducibility.

We believe these revisions have significantly improved the quality of our manuscript and we hope it now meets the standards of your esteemed journal. We look forward to your feedback and are happy to address any further questions or concerns.

Sincerely,

Shinya Mizuno

Juntendo University, Faculty of Health Data Science

6-8-1, Hinode, Urayasu City, Chiba 279-0013, Japan

---

## [Decision Letter · Decision Letter 1]

20 Sep 2024

Limitations of Calculating Theoretical Solutions for Closed BCMP Queueing Networks and Verification of Alternative Theoretical Values by Parallel Simulation

PONE-D-24-06423R1

Dear Dr. Mizuno,

We’re pleased to inform you that your manuscript has been judged scientifically suitable for publication and will be formally accepted for publication once it meets all outstanding technical requirements.

Kind regards,

Palaniyappan Sathyaprakash, Ph.D

Academic Editor

PLOS ONE

Additional Editor Comments (optional):

Reviewers' comments:

Reviewer's Responses to Questions

**Comments to the Author**

1. If the authors have adequately addressed your comments raised in a previous round of review and you feel that this manuscript is now acceptable for publication, you may indicate that here to bypass the “Comments to the Author” section, enter your conflict of interest statement in the “Confidential to Editor” section, and submit your "Accept" recommendation.

Reviewer #1: All comments have been addressed

Reviewer #2: All comments have been addressed

2. Is the manuscript technically sound, and do the data support the conclusions?

Reviewer #1: Yes

Reviewer #2: Yes

3. Has the statistical analysis been performed appropriately and rigorously? 

Reviewer #1: Yes

Reviewer #2: Yes

4. Have the authors made all data underlying the findings in their manuscript fully available?

Reviewer #1: Yes

Reviewer #2: Yes

5. Is the manuscript presented in an intelligible fashion and written in standard English?

Reviewer #1: Yes

Reviewer #2: Yes

6. Review Comments to the Author

Reviewer #1: Thank you for your resubmission. After reviewing the revised version of the manuscript, I am pleased to see that the authors have carefully addressed all the comments and suggestions provided in the previous round of review.

Addressing Reviewer Feedback: The authors have responded comprehensively to the feedback provided, and the changes made to the manuscript have significantly improved its quality. The clarity of the arguments, the methodology, and the results discussion have been enhanced, aligning with the suggestions.

Clarity and Coherence: The revisions to the introduction and discussion sections have improved the manuscript's coherence, particularly in terms of context and contribution to the field. The restructuring has resulted in a more logical flow of ideas.

Methodological Improvements: The adjustments made to the methodology have satisfactorily resolved the concerns raised previously. The additional details provided make the approach more transparent and reproducible.

Presentation and Writing Style: The quality of the writing has been improved, with clearer expression of ideas, corrected grammatical issues, and well-organized sections. These changes have enhanced the readability of the manuscript.

Figures and Tables: The modifications to the figures and tables are effective and provide a clearer representation of the data, as requested.

In conclusion, the authors have successfully addressed all concerns raised in the previous review, and the manuscript is now ready for publication in its current form. I recommend acceptance of the paper.

Reviewer #2: The author(s) has addressed all concerns I raised and have improved the paper largely. in fact, the results presented are interesting and bring novelty the topic presented in the paper. This paper is publishable.

7. PLOS authors have the option to publish the peer review history of their article (what does this mean?). If published, this will include your full peer review and any attached files.

Reviewer #1: **Yes: **dr. upasana dohare

Reviewer #2: No

---

## [Editor Report · Acceptance letter]

24 Sep 2024

PONE-D-24-06423R1 

PLOS ONE

Dear Dr. Mizuno, 

I'm pleased to inform you that your manuscript has been deemed suitable for publication in PLOS ONE. Congratulations! Your manuscript is now being handed over to our production team.

Kind regards, 

on behalf of

Dr. Palaniyappan Sathyaprakash 

Academic Editor

PLOS ONE